# Nonspecific Orbital Inflammation (NSOI): Unraveling the Molecular Pathogenesis, Diagnostic Modalities, and Therapeutic Interventions

**DOI:** 10.3390/ijms25031553

**Published:** 2024-01-26

**Authors:** Kevin Y. Wu, Merve Kulbay, Patrick Daigle, Bich H. Nguyen, Simon D. Tran

**Affiliations:** 1Department of Surgery, Division of Ophthalmology, University of Sherbrooke, Sherbrooke, QC J1G 2E8, Canada; yang.wu@usherbrooke.ca (K.Y.W.);; 2Department of Ophthalmology & Visual Sciences, McGill University, Montreal, QC H4A 0A4, Canada; 3CHU Sainte Justine Hospital, Montreal, QC H3T 1C5, Canada; 4Faculty of Dental Medicine and Oral Health Sciences, McGill University, Montreal, QC H3A 1G1, Canada

**Keywords:** autoimmunity, autoimmune diseases, immune system dysregulation, genetic predisposition, epigenetic influences, molecular pathogenesis, nonspecific orbital inflammation, signaling pathways, cytokines

## Abstract

Nonspecific orbital inflammation (NSOI), colloquially known as orbital pseudotumor, sometimes presents a diagnostic and therapeutic challenge in ophthalmology. This review aims to dissect NSOI through a molecular lens, offering a comprehensive overview of its pathogenesis, clinical presentation, diagnostic methods, and management strategies. The article delves into the underpinnings of NSOI, examining immunological and environmental factors alongside intricate molecular mechanisms involving signaling pathways, cytokines, and mediators. Special emphasis is placed on emerging molecular discoveries and approaches, highlighting the significance of understanding molecular mechanisms in NSOI for the development of novel diagnostic and therapeutic tools. Various diagnostic modalities are scrutinized for their utility and limitations. Therapeutic interventions encompass medical treatments with corticosteroids and immunomodulatory agents, all discussed in light of current molecular understanding. More importantly, this review offers a novel molecular perspective on NSOI, dissecting its pathogenesis and management with an emphasis on the latest molecular discoveries. It introduces an integrated approach combining advanced molecular diagnostics with current clinical assessments and explores emerging targeted therapies. By synthesizing these facets, the review aims to inform clinicians and researchers alike, paving the way for molecularly informed, precision-based strategies for managing NSOI.

## 1. Introduction

The orbit, a region containing some of the most sophisticated anatomies in the human body, is host to multiple structures whose complexity can give rise to various pathologies. These conditions can profoundly impact a patient’s vision, aesthetic confidence, overall well-being, and quality of life. Among these, non-specific orbital inflammation (NSOI), while generally benign, can present unique challenges. NSOI, colloquially known as orbital pseudotumor, represents 8% to 11% of all orbital masses and is particularly prevalent in adults, especially middle-aged females. Cases of NSOI may lead to relapse and refractory symptoms, causing considerable distress and complicating patient management despite their typically benign nature. This complexity of NSOI, with its idiopathic nature, presents significant challenges for clinicians. This article aims to deepen the understanding of NSOI’s pathogenesis and pathophysiology, highlighting the need for molecular insights to develop treatments like biologics and immunomodulators. Emphasizing the role of bench science, it connects molecular research with clinical advancements, aiming to enhance patient outcomes.

The synthesis of knowledge underlines the importance of a multidisciplinary approach to understanding and managing NSOI. The review conducts an extensive literature analysis over the past nine years, focusing on NSOI’s molecular aspects. This approach highlights the significance of fundamental science research in contributing to clinical care and patient well-being, bridging the gap between molecular insights and practical therapeutic applications for NSOI.

## 2. Etiology and Pathogenesis

### 2.1. Etiologies of Nonspecific Orbital Inflammation

The triggers involved in NSOI pathogenesis remain to be fully elucidated. Although the term NSOI is attributed to orbital inflammation cases where a clear etiology cannot be identified, clinical associations with infections, autoimmune and systemic diseases, pharmaceutical drugs, environmental factors, and individual characteristics have been reported (Figure 1) [1]. It is important to remember that in some cases, no clinical association can be made [2,3]; therefore, NSOI is always a diagnosis of exclusion.

#### 2.1.1. Autoimmune and Systemic Diseases

NSOI has been shown to occur in association with diverse autoimmune conditions. The main hypothesis suggests a genetic predisposition or a dysregulation in the immune system—mainly in the function of T cells—where concerned individuals are more prone to NSOI occurrence in the context of autoimmune diseases [4]. Thyroid eye disease (TED), regardless of thyroid function status, was shown to be associated with NSOI in case reports [5,6,7,8,9]. In fact, TED is the most commonly associated disease with NSOI. Other conditions linked to NSOI are Crohn’s disease [10,11,12], psoriasis [13], discoid lupus [14,15,16], Behçet disease [17,18], sarcoidosis [19], Wegener’s granulomatosis [20], Churg-Strauss syndrome [21,22], and giant cell arthritis [23,24,25]. It is important to remember that NSOI may present with other connective tissue diseases, and it should always remain on the list of differentials until another cause can be identified.

#### 2.1.2. Viral Infections

Conversely, systemic infections have also shown an association with the incidence of NSOI, such as upper respiratory tract infections [26,27], paranasal sinusitis [28,29], and vaccination (e.g., influenza, COVID) [30,31,32]. Dacryoadenitis, a possible clinical manifestation of Epstein-Barr virus (EBV), herpes simplex virus (HSV), and SARS-CoV-2 infection [33,34,35,36], can lead to NSOI due to failure of corticosteroid treatment. Similarly, the human immunodeficiency virus (HIV) can lead to the development of orbital myositis in immunosuppressed patients through a process called immune reconstitution inflammatory syndrome, where T cells mediate a proinflammatory phenomenon [37]. Although numerous viral causes are mentioned in this section, EBV infections are the most reported association with NSOI.

In the context of NSOI, EBV causes severe lymphocytic infiltration [38]. Ren et al. have studied the histopathological features of NSOI-biopsied tissues. In NSOI tissues, lymphocytic infiltration was found in 57.10% of samples, whereas fibrotic changes were only reported in 12.50% of samples [39]. Furthermore, EBV-encoded small RNAs (EBERs) were found to be expressed in NSOI tissues [34,39,40]. However, a prior study demonstrated the absence of EBV-VCA-IgM in the presence of EBV-VCA-IgM in patients with NSOI, suggesting a past history of EBV infection (i.e., no acute infections were shown) [40]. Furthermore, EBV-DNA was also found in control groups [40]. However, a more recent study demonstrated the lack of EBV-DNA in blood samples of patients with NSOI, whereas it was shown to be present in approximately 40% of patients with TED and control groups [41]. Overall, given the sparse literature on EBV-induced NSOI, associations between both disease entities have to be interpreted judiciously. 

#### 2.1.3. Environmental Factors

A case-control study from the Netherlands, with the primary objective of identifying the risk factors for NSOI occurrence, has shown an increased incidence of NSOI in patients with a higher BMI (odds ratio (OR) 2.88, 95% CI 1.32 to 6.32) [4]. In fact, obesity is known to induce chronic inflammation through an increased accumulation of immune cells in adipose tissue, mainly macrophages and T cells [42]. Having shown an increase in T cell expression in idiopathic orbital inflammatory pseudotumor tissues, it is hypothesized that chronic low-grade inflammation could trigger NSOI through immune imbalance [38]. Conversely, the risk of NSOI was shown to be decreased in women with an advanced age at first childbirth (OR 0.14, 95% CI 0.03 to 0.64) and in patients with a higher socioeconomic status (OR 0.38, 95% CI 0.17 to 0.84) [4].

An association between certain drugs (e.g., bisphosphonates, lithium, and chemotherapy) and NSOI has also been outlined in the literature [2]. Bisphosphonates, also known as osteoclast-inhibiting agents, are mainly used for bone-resorbing clinical conditions such as osteoporosis [43]. The association of aminobiphosphonates with NSOI could be attributable to their structural homology with the ligand socket of γδ T cells, therefore leading to T cell activation and inflammatory mediator release [44,45,46,47]. 

### 2.2. Pathogenesis 

Previously, we have covered several of the etiologies of this clinical entity, where they all have one common factor: a dysregulation of the immune system. A large portion of our current understanding of NSOI comes from the immunohistochemical analysis of the inflammatory orbital tissue and flow cytometer analysis of the patient’s plasma. The hallmarks of NSOI involve a nonspecific, polymorphous infiltrate of the orbital tissues by B- and T-cells, neutrophils, eosinophil granulocytes, histiocytes, and macrophages [48]. On immunohistopathology analysis, focally organized zones can be distinguished, corresponding to lymphoid follicles with reactive germinal centers as well as fibrosis within the connective tissues [48]. Amongst the first evidence regarding an autoimmune process in NSOI, Atabay et al. demonstrated in 1995 the presence of antibodies directed towards the eye muscle membrane antigens in patients with NSOI [49]. Several years later, the presence of toll-like receptors (TLRs) in biopsies from patients with NSOI was demonstrated, mainly TLR-2, -3, and -4 [50]. TLRs are involved in the recognition of pathogen-associated molecular patterns (PAMM) and can be either localized on the cell surface (e.g., TLR-2 and -4) or within the intracellular compartments (e.g., TLR-3) [51]. Upon activation, they initiate the innate immune system and can form a bridge with the adaptative immune system. Since these descriptive discoveries, multiple preclinical studies have sought to determine the function of the innate and adaptative immune systems in the pathogenesis of NSOI by proposing novel hypotheses [38]. 

#### 2.2.1. Overview of the Innate and Adaptative Immune Systems

Prior to covering the molecular dysregulations involved in the pathogenesis of NSOI, a better understanding of the innate and adaptative immune systems is required (Figure 2). 

The innate immune system represents the first line of defense of the body and is composed of anatomic, physiologic, phagocytic, and inflammatory barriers. The molecular signaling pathways of the innate immune system are activated by the pattern recognition receptors (PRRs), which are designed to recognize pathogen-associated molecular patterns (PAMPs) [52]. TLRs are among the most studied types of PRRs; they are type 1 transmembrane glycoproteins and encompass three regions: an extracellular, intracellular, and transmembrane region [53]. The extracellular region harbors leucine-rich regions (LRRs) and is involved in the pattern recognition ability of these receptors. PAMPs, such as nucleic acids (e.g., DNA, double-stranded RNA (dsRNA), single-stranded RNA (ssRNA)), surface glycoproteins, lipoproteins, and membrane components, are widely known activators of TLRs [54]. Upon ligand binding, TLRs trigger intracellular signaling cascades involved in the expression of proinflammatory molecular partners. TLR-2, -3, and -4 have been shown to be involved in the pathogenesis of NSOI [50]. TLR-2 signaling occurs following receptor heterodimerization with TLR-1 or TLR-6, with subsequent MyD-88 intracellular signaling pathway activation [55]. The MyD-88 cascade ultimately leads to the nuclear translocation of nuclear factor κ B (NFκB) and the activation of the serine/threonine-specific mitogen-activated protein kinase (MAPK) p-38 [55]. The function of the NFκB transcription factor in inflammation has been thoroughly reviewed previously by Liu and colleagues [56]. It is mainly involved in proinflammatory cytokine production (e.g., interleukin (IL)-1, 2, -6, -8, -12, and tumor necrosis factor α (TNFα)), chemokine production, proinflammatory gene induction, and T cell activation and differentiation through IL-12 and -13 production [56]. Similarly to TLR-2, TLR-4 mediates its signaling pathway through the recruitment of MyD-88 and is a known sensor for bacterial lipopolysaccharides (LPS) [57]. Although not covered in this review, TLR-4 also possesses a MyD-88-independent signaling pathway that is activated upon receptor internalization, which ultimately leads to type 1 interferon (IFN) production [57]. TLR-4 was shown to be involved in acute and chronic inflammatory disorders [58]. Conversely, TLR-3, an intracellular TLR, leads to the activation of the MAPK signaling pathway and the transcription factors NFκB, the activator protein 1 (AP-1), and IRF3/7, which mediate the production of interferons (IFNs), proinflammatory cytokines, and chemokines [59]. Once the proinflammatory cascades are initiated in conjunction with antigen-presenting cells (APCs), the proinflammatory microenvironment leads to the activation of T and B cells. APCs are the key molecular partners involved in activating the adaptative immune system, where cytokines play a crucial role. In fact, following T cell receptor (TCR) activation through APCs, naïve T CD4^+^ cells differentiate into different Th lineages according to the expression and combination of different cytokines (Figure 3) [60].

#### 2.2.2. Dysregulations in the Immune System

Dysregulations in the immune system constitute the mainstay of NSOI pathogenesis (Figure 4). The importance of the inflammatory microenvironment in autoimmune and systemic diseases is well known [61,62]. To investigate the biomolecular changes involved in the NSOI microenvironment, quantitative cytokine assays were performed in patients with NSOI [63]. It was shown that IL-2, -8, -10, -12, IFN-γ, and TNFα expressions were significantly elevated in NSOI, with greater levels of increase in IL-12 and IFN-γ [63]. These two latter cytokines are tightly involved in the expression of Th1 cells; IL-12 promotes T cell differentiation into the Th1 subtype, whereas IFN-γ is mainly secreted by activated Th1 cells [64,65]. IL-12 induces Th1 differentiation by activating the transcription factor STAT3 (i.e., signal transducer and activator of transcription 3). Once polarized into the Th1 subset, Th1 cells induce IFN-γ secretion and express the chemokine receptor CXCR3, which is a chemoattractant for T cells, neutrophils, and macrophages [64,66]. The role of Th1 cells in autoimmune diseases has been widely reviewed; Th1 cells drive chronic autoimmune responses [67]. Similarly, surface markers of APCs and activated T cells (e.g., HLA-DRB_1_ and HLA-DQ_1_) were shown to be significantly increased in NSOI, therefore implying a crucial role of the Th1-mediated immune response in the pathogenesis of NSOI [68]. Given these results, an upregulation in the expression of Th1 T cells is hypothesized to occur.

Regulatory T cells (Tregs; CD4^+^CD25^+^) are key inflammatory response regulators and hold a crucial role in immune homeostasis; poor Treg cell function is linked with autoimmune diseases [69]. With this in mind, Chen et al. have recently demonstrated an increase in dysfunctional Tregs in the peripheral blood of patients with NSOI [70]. Tregs possess regulatory functions where they limit responsiveness to self-antigens in order to suppress overactivation of immune responses [71,72]. Polarization of Tregs through specific transcription factor expression acquisition can lead to their differentiation into Treg subsets: Th1-like, Th2-like, or Th17-like cells. Th1 cells are mainly involved in cell-mediated responses through IL-2 and IFN-γ secretion, whereas Th2 cells mediate humoral immune responses through IL-4, -5, and -13 secretion [73]. Th17 cells mediate host defense mechanisms and have been shown to play a role in many autoimmune diseases [74]. In patients with NSOI, circulating Tregs were shown to be polarized into a Th17-like phenotype, whereas Tregs isolated from the orbital tissues were of Th2-like phenotypes [70]. T CD4+ cell differentiation into Th17-like phenotypes is achievable in the presence of tumor growth factor (TGF)-β and IL-6, whereas differentiation into Th2-like phenotypes is dependent on IL-4 [75,76,77]. Orbital Tregs have a dysregulated function: suppressive capacity for naïve T cell proliferation was shown to be decreased, which was associated with an increased production of IL-4 by conventional T cells (Tconvs) [70]. Furthermore, the expression of the chemokine CCL17—also known as the ligand of the chemokine receptor CCR4—was enhanced in collected tissues and was shown to promote circulating Tregs chemotaxis to the orbital tissue [70]. The increase of the Th2-like phenotype in the orbital tissue of patients with NSOI was shown to enhance tissue fibrosis due to a downregulation in the IL-33/ST2 signaling pathway and an increase in CD40^+^ fibrocytes in the inflammatory orbital tissue [70,78]. IL-33 is known to enhance the expression of ST2 in Tregs, which positively correlates with the anti-inflammatory actions of Tregs [79,80]. It was shown that IL-33 mRNA expression in the orbital tissues of patients with NSOI was significantly decreased [70], which supports the observed pro-inflammatory and fibrotic effects of NSOI. Conversely, in vitro treatment of NSOI-derived Tregs with IL-33 was shown to suppress the proinflammatory and profibrotic actions of these cells through IFN-γ and orbital fibroblast activation downregulation [70]. Supporting the evidence of increased fibrosis in orbital adipose tissue, fibrosis-related transcripts (e.g., lumican, fibronectin, collagens type I and VIII, and thrombospondin) were shown to be significantly upregulated in NSOI samples [81]. Moreover, CD40^+^ fibrocytes within the orbital inflammatory tissues were shown to secrete IL-6 [78]. IL-6 plays a crucial role in chronic autoimmune and inflammatory diseases; it promotes a proinflammatory microenvironment through the stimulation of acute-phase protein production [82]. In addition, IL-6 is a direct modulator of plasma cells, where it promotes antibody production through dendritic cell (DC) maturation and the STAT3 signaling pathway [77]. 

Dendritic cells (DCs) are the key players involved in the crosstalk of the innate immune response with the adaptative immune response. They can be divided into two categories: plasmacytoid DCs (pDCs) and conventional DCs (cDCs). cDCs mainly act as APCs, whereas pDCs regulate B cell differentiation and immunoglobulin secretion [83]. To evaluate the cell composition in peripheral blood of patients with NSOI, mainly DCs, given the presence of germinal centers in histopathology analysis of NSOI biopsies, Laban et al. performed a multiparametric flow cytometer analysis of meta-clusters containing DCs [84]. It was shown that patients with NSOI exhibit decreased levels of pDCs as well as cDCs type 2 [84]. Given that pDCs are involved in B cell differentiation and subsequent immunoglobulin secretion, it was hypothesized that a decrease in their expression in idiopathic orbital inflammatory biopsy tissues, with concomitant B cell infiltration, could be the result of a negative feedback loop by self-maintained B cell expansion [84,85]. These results could explain the reported presence of anti-eye muscle membrane antigen antibodies in patients with NSOI [49].

#### 2.2.3. The Role of miRNAs 

More recently, the role of small noncoding regulatory RNAs (miRNAs) in the pathogenesis of NSOI has been investigated. miRNAs act by binding to the 3′ untranslated region (UTR) of the target mRNA, therefore inducing repression in translation processes [86,87]. miRNA binding to the 5′ UTR has also been reported [88], as have interactions with the promoter region that induce gene transcription [89]. Over the past years, miRNAs have been extensively studied for their role in inflammation and immune regulation [90,91]. By performing OpenArray miRNA profiling in patients with NSOI, a miRNA cluster was shown to be significantly increased in these patients; the cluster contained miR-140-5p, miR-148a-3p, miR-193a-5p, miR-223-3p, miR-223-5p, miR-29a-3p, miR-365a-3p, and U6 snRNA [92]. However, further studies are required to better understand the hypothesized pathogenic role of these dysregulations (Table 1). 

miR-223 is a known regulator of immune cell differentiation and inflammation; during inflammatory processes, miR-223 is upregulated in granulocytes, macrophages, and T cells [93]. It is involved in granulocyte and macrophage proliferation and differentiation, as well as in DC-related functions [93]. MiR-223 was also shown to induce Treg differentiation in vivo [94]. Furthermore, nuclear factor κ B (NFκB) was shown to upregulate miR-223 expression in Jurkat T cells [95]. Interestingly, genes involved in the NFκB and PI3K-AKT signaling pathways were shown to be enhanced in patients with NSOI [68]. Furthermore, we previously discussed the alterations in Treg expression and function in NSOI, which are the key actors in this craft of inflammation. miR-223-3p and miR-223-5p upregulation could therefore be key molecular partners in promoting chronic inflammation.

Cytokine regulation by miR-148a was also reported to be involved in autoimmunity [96]. miR-148a was shown to be involved in monocyte-derived DCs by directly targeting the *MAFB* mRNA—a key regulator of hematopoiesis [96]. Furthermore, miR-148a was shown to soothe the progression and development of psoriasis in a mouse model [96]. The role of miR-148a in the establishment of chronic inflammation in NSOI can be attributable to a dysregulation of B cell tolerance. In fact, increased expression of miR-148a was shown to favor lethal autoimmunity in a lupus mouse model by impairing the B cell tolerance mechanism through suppression of *Gadd45a*, *Pten*, and *Bcl2l11*, key encoders of the pro-apoptotic factor Bim [97,98]. These observed molecular pathways can be transposed to the pathogenesis of NSOI, where autoantibodies were previously noted in affected patients.

In a murine asthmatic model, miR-365-3p was shown to negatively regulate IL-17-induced inflammatory cytokines by targeting ARRB2 [99]. IL-17, mainly produced by the Th17 cell lineage, is known to initiate potent inflammatory pathways through the induction of neutrophil-specific chemokines, IL-6, IL-1, and TNFα production [103,104]. Furthermore, in the synoviocytes of mice with rheumatoid arthritis (RA), miR-365-3p was shown to accelerate apoptosis and inhibit cell proliferation by downregulating the expression of IL-1β and IL-6 [100]. Similar actions of miRNA-140 were also reported; an abnormal increase in miRNA-140 expression leads to cell death by apoptosis [101]. Moreover, miR-193a was also shown to play a role in innate immunity; miR-193a promotes granulocyte differentiation [102]. Overall, the increase in specific miRNA in the context of NSOI could be linked to the proinflammatory microenvironment. 

Findings regarding the role of miR-29a-3p and U6 snRNA in chronic inflammation, especially in the context of NSOI, still need further investigation. Tokić and colleagues have shown a downregulation in miR-29a-3p in patients with Hashimoto’s thyroiditis (HT), associated with an increase in T-bet mRNA [105]. Given that T-bet is a key regulator of T cell differentiation and that the expression of miR-29a-3p was shown to be increased in patients with NSOI, its specific role in inflammation and autoimmunity in this disease remains to be elicited [106].

## 3. Clinical Overview

NSOI, encompassing a spectrum of terminologies like orbital pseudotumor, orbital inflammatory pseudotumors (OIP), idiopathic orbital inflammation (IOI), orbital inflammatory syndrome (OIS), and nonspecific orbital inflammatory pseudotumor (NSOIP), is a non-neoplastic, space-occupying orbital disorder characterized by inflammation without an identifiable infectious, systemic, or malignant cause [107]. Ranking as the third most prevalent orbital condition in adults, following thyroid orbitopathy and orbital lymphoma, NSOI typically has no signs or evidence of an underlying systemic infection or neoplastic process [108]. However, clinically and radiologically, NSOI often presents similarities to malignant conditions. Consequently, NSOI is identified through a process of exclusion, established only after all underlying causes have been thoroughly investigated and ruled out [109]. 

NSOI exhibits variable clinical presentations, ranging from localized to diffuse involvement. In its localized form, NSOI specifically targets areas such as the extraocular muscles, leading to orbital myositis, and the lacrimal gland, resulting in dacryoadenitis. Additionally, inflammation can be observed in the sclera (scleritis), uvea (uveitis), and encompassing the superior orbital fissure and cavernous sinus, characteristic of Tolosa–Hunt syndrome [110]. Other manifestations include periscleritis, perineuritis, and isolated orbital masses. In contrast, the diffuse variant of NSOI is characterized by widespread involvement of the orbital fatty tissues [109].

Pediatric NSOI is relatively rare but tends to be bilateral and frequently accompanied by systemic signs. These signs include headache, vomiting, loss of appetite, fatigue, and fever, which are observed in approximately half of the pediatric patients, and there is a notable correlation with ocular conditions such as iritis, uveitis, and optic disc edema [111]. Laboratory findings often show elevated erythrocyte sedimentation rate (ESR) and eosinophilia, which can aid in diagnosis [111]. Despite these symptoms, pediatric NSOI generally carries a lower risk of underlying systemic diseases compared to adult NSOI. 

### 3.1. Epidemiology

Orbital pseudotumor, representing 8% to 11% of all orbital tumors, is more prevalent in adults, particularly middle-aged females, and has a global presence [107,108]. Studies have estimated its incidence among orbital disorders to be between 6% and 16% [112]. However, the true incidence of NSOI is challenging to determine due to its varied manifestations and the absence of a universally accepted definition. It predominantly affects the lacrimal gland and typically presents at 30 to 60 years of age [107]. In adults, the occurrence is typically unilateral; however, bilateral occurrences are more frequent in children. Pediatric cases also exhibit a high recurrence rate of up to 76%, compared to the recurrence rate in the adult population, which ranges from 33% to 58% [113]. In a study by Swamy et al. [114], biopsy-proven NSOI was analyzed in 24 patients, revealing that the lacrimal gland was affected in 54.2% of cases (13/24), extraocular muscles in 50.0% (12/24), orbital fat in 75.0% (18/24), sclera in 4.2% (1/24), optic nerve in 20.8% (5/24), and other areas in 8.3% (2/27).

### 3.2. Histopathological Considerations

In the context of NSOI, the role of orbital biopsy is a subject of ongoing debate. Current treatment strategies frequently employ systemic corticosteroids, with the recommendation for biopsy primarily in scenarios where patients exhibit inadequate or partial responses to such steroid treatments [110,115]. Additionally, for individuals with a past record of systemic malignancies or where there is continued ambiguity in diagnosis, the implementation of a biopsy is strongly advised [116].

Despite this prevailing uncertainty, an alternative viewpoint promotes the utilization of biopsy, highlighting its relatively low morbidity. This school of thought considers the possibility that other orbital conditions may also respond to corticosteroid treatment and the significant incidence of systemic diseases affecting the lacrimal gland. As a result, numerous specialists recommend diagnostic biopsy for all nonmyositic lesions, especially those unconnected to the optic nerve [116]. Furthermore, the surgical excision of nonspecific dacryoadenitis during the biopsy could yield both diagnostic and therapeutic advantages.

On the other hand, in cases involving myositis or lesions affiliated with the optic nerve or orbital apex, the presence of unique clinical and radiographic indicators often strongly supports a diagnosis. Here, the risk of performing a biopsy might outweigh its benefits, taking into account the procedure’s inherent risks and the high probability of an accurate diagnosis through non-invasive methods.

Achieving a confirmatory diagnosis of NSOI is predominantly reliant on histopathological analysis. The gold standard in diagnostic procedures for NSOI includes fine-needle aspiration biopsy or incisional/excisional biopsy techniques. While fine-needle aspiration is less invasive and preferable in some cases, it is important to note that the typically firm consistency of the tumor often results in suboptimal yields from these biopsies [117]. 

Histologically, NSOI is characterized by a heterogeneous mix of cellular infiltrates, mainly consisting of lymphocytes, plasma cells, and eosinophils, alongside varying degrees of reactive fibrosis [107,110]. Notably, the sclerosing variant of NSOI shows a predominance of fibrosis with sparse cellular inflammation. Distinguishing this subtype is crucial, as it presents differently from hypercellular lymphoid proliferations, which are separate clinical and histological entities. Furthermore, recent advancements in immunostaining methodologies have significantly enhanced the capability to differentiate between lymphoma and pseudotumor, thereby improving both the specificity and sensitivity of the histological diagnosis of these conditions [117]. Immunohistochemical analysis for IgG4 in plasma cells is also crucial to exclude IgG4-related disease (IgG4-RD), as tissue plasma cell IgG4 positivity is not commonly observed in NSOI.

In the context of NSOI, the presence of caseating granulomatous inflammation or vasculitis generally suggests an alternative diagnosis [48]. Yet, it is noteworthy that there is a unique granulomatous subtype within NSOI that closely resembles sarcoidosis. This variant is characterized by histiocytic infiltration and the development of well-defined noncaseating granulomas [107,110].

### 3.3. Diagnostic Approach and Clinical Manifestations

The identification of orbital pseudotumor hinges on an exclusionary diagnostic approach, necessitating comprehensive medical history analysis to distinguish it from systemic diseases with overlapping clinical features. These conditions include, but are not limited to, sarcoidosis, granulomatosis with polyangiitis, Sjogren’s syndrome, IgG4-related disease (IgG4-RD), lymphoproliferative and histiocytic disorders, xanthogranulomatous diseases, or metastases [48]. In the absence of pertinent historical clues, additional laboratory workups are imperative to conclusively eliminate the likelihood of these mimickers.

In the differential diagnosis, several categories warrant consideration:Neoplastic processes, particularly metastatic involvement and primary orbital neoplasms;Hematological malignancies, such as lymphoma and leukemia;Inflammatory and autoimmune disorders, notably IgG4-related disease, TED, sarcoidosis, Sjogren’s syndrome, and granulomatosis with polyangiitis (GPA);Infectious etiologies, with orbital cellulitis as a prime example. Other infectious causes, like syphilis and tuberculosis, are crucial to be considered as part of the differential diagnosis.

The clinical presentation of NSOI encompasses a broad array of symptoms, with cases displaying anything from widespread orbital inflammatory signs to localized afflictions of specific orbital structures, including the lacrimal gland and extraocular muscles [118]. The temporal onset of NSOI is variable, manifesting in forms that range from acute to subacute, with the possibility of evolving into chronic conditions or demonstrating a pattern of relapse [110,119]. 

Typically, patients may present with eyelid erythema and edema, as well as some degree of ptosis. This is frequently accompanied by conjunctival erythema and chemosis [108]. A hallmark feature is a deep, boring pain that intensifies with extraocular muscle movement, signaling possible extraocular muscle involvement and restriction. The inflammation of these muscles can lead to restrictive ophthalmoplegia and subsequent diplopia [111] (Figure 5). The onset of proptosis may be rapid or occur over time, contributing to visual impairment. Such visual deficits may stem from exposure keratopathy linked to significant proptosis, from optic nerve compression due to a mass effect at the orbital apex, or in the context of compartment syndrome. Alternatively, vision loss could be related to posterior scleritis, potentially involving exudative retinal detachment [108,111,120]. In asymptomatic or mildly symptomatic individuals, imaging modalities may inadvertently reveal the presence of an orbital mass, thus aiding in the diagnosis of NSOI [108,111,120].

Accounting for half of NSOI occurrences, dacryoadenitis typically manifests as a painful, palpable mass situated in the lateral aspect of the upper eyelid, which may induce an S-shaped ptosis and can present bilaterally [110]. Clinicians should be cognizant of the potential for NSOI-related dacryoadenitis to mimic or coincide with IgG4-related ophthalmic disease (IgG4-ROD). This resemblance is crucial to consider, as IgG4-ROD similarly affects the orbital soft tissue and may present with comparable clinical features [110]. However, the cardinal signs of inflammation are often more prominent with NSOI than with IgG4-ROD. Pain and redness can be minimal in IgG4-ROD, and presentation may be limited to proptosis or periocular masses.

NSOI myositis and TED myositis share several demographic and clinical similarities but also exhibit distinct differences. Both conditions predominantly affect a young to middle-aged female population and present with similar clinical manifestations, including signs of orbital inflammation and enlargement of the extraocular muscles, detectable through radiographic imaging. Additionally, patients with either NSOI myositis or TED myositis generally respond favorably to corticosteroid therapy. However, a distinguishing feature between the two is the involvement of the muscle tendon; TED myositis typically spares the muscle tendons, whereas NSOI myositis tends to involve them. Furthermore, there is a notable difference in the pattern of extraocular muscle involvement. In NSOI myositis, the most commonly affected muscle is the medial rectus, followed by the superior, lateral, and inferior recti muscles. In contrast, TED myositis frequently affects the muscles in a different order, with the inferior rectus being most commonly involved, followed by the medial, superior, and lateral recti muscles [110]. A careful assessment of the upper lid position can also help differentiate between the two conditions. NSOI is commonly associated with ptosis, whereas TED typically leads to retraction.

### 3.4. Diagnostic Tools

#### 3.4.1. Laboratory Testing

The diagnostic process for NSOI is multifaceted, involving an array of laboratory tests and imaging studies. The primary aim of these laboratory tests is more to rule out other diseases than to confirm the diagnosis of NSOI. Given the known association between rheumatologic diseases and NSOI, the standard laboratory workup for suspected NSOI is extensive. It should include a complete blood count, a basic metabolic panel, thyroid function tests, erythrocyte sedimentation rate (ESR), antinuclear antibodies (ANA), antineutrophil cytoplasmic antibodies (ANCA), angiotensin-converting enzyme (ACE) level, rapid plasma reagin (RPR) test, and rheumatoid factor (RF) [111]. Additionally, screening for infectious causes like syphilis and tuberculosis is crucial to exclude these etiologies.

#### 3.4.2. Imaging Techniques

The role of imaging in NSOI is to complement clinical assessment as well as gauge the response to therapeutic interventions. Orbital B-scans (i.e., ultrasound) can be particularly useful in detecting ocular complications like exudative retinal and choroidal detachments [116]. 

CT and MRI scans are instrumental in identifying the morphology and extent of NSOI. CT is particularly useful for assessing the orbital bones and sinuses [121], while MRI excels at depicting soft tissue changes, especially in areas like the cavernous sinus and superior orbital fissure [122]. Kapur et al. [123] have highlighted the utility of diffusion-weighted imaging (DWI) in differentiating NSOI from conditions like orbital cellulitis and orbital lymphoma based on distinct intensity patterns [123]. These radiologic findings facilitate a more precise classification of NSOI subtypes:Lacrimal Gland: Typically shows diffuse enlargement while retaining shape, with notable blurring and lateral expansion;Extraocular Muscles: Frequent enlargement of single or multiple muscles, often involving the medial rectus, with tendon enlargement, differing from thyroid orbitopathy’s tendon sparing;Optic Nerve: Presence of the “tramline” sign, indicative of surrounding inflammation;Sclera and Related Structures: Generally, exhibit nonspecific thickening with occasional blurred scleral margins;Orbital Fat: Characterized by diffuse inflammation, potentially encasing the globe and optic nerve sheath;Orbital Apex and Intracranial Areas: NSOI may cause optic nerve compression and extend into the cavernous sinus and middle cranial fossa, with associated tissue changes.

These imaging features are integral in diagnosing NSOI and determining its extent, thereby guiding appropriate management strategies.

### 3.5. Therapeutic Approaches for Nonspecific Orbital Inflammation

The management of nonspecific orbital inflammation encompasses various therapeutic modalities, including pharmacological interventions and radiation therapy (Table 2). Primary treatment options include:Corticosteroids: These are the cornerstones of therapy, often resulting in rapid symptom resolution. In cases of NSOI, a dramatic response is typically observed within 48 h following systemic prednisolone administration. Local corticosteroid injections are also an alternative [118,124,125,126];Radiation Therapy: Employed in unresponsive cases, radiotherapy uses a low dose of 20 to 30 Gy, delivered in fractions of 2 Gy each [124];Immunosuppressive Agents: These agents are particularly beneficial for patients who exhibit either non-responsiveness or recurrence post-corticosteroid therapy [110]. These include the following:Methotrexate;Cyclosporin-A;Mycophenolate mofetil;Cyclophosphamide.Biological Agents: These are typically reserved for recalcitrant cases of nonspecific orbital inflammation [110]. These encompass the following: Infliximab;Adalimumab;Etanercept;Daclizumab;Abatacept;Tocilizumab;Rituximab.

The corticosteroid treatment, while effective for many, has a limited cure rate of 37% and a recurrence rate of 52%. Long-term steroid use is associated with systemic side effects like insomnia, hyperglycemia, weight gain, and cataracts [126]. Therefore, for recurrent disease, corticosteroid-sparing therapy, including immunosuppressive and immunomodulatory therapies, should be considered [111].

The molecular mechanisms underpinning the efficacy of these biologics are intricate and are discussed in detail in a later section of this review article. This section will provide a comprehensive understanding of how these agents interact at the molecular level to mediate their therapeutic effects in nonspecific orbital inflammation.

Despite being benign, nonspecific orbital inflammation can have a clinically fulminant course, potentially leading to vision loss and significant oculomotor dysfunction. This necessitates prompt and aggressive treatment, especially in cases involving optic nerve impairment, to prevent long-term visual consequences [120].

**Table 2 ijms-25-01553-t002:** Summary of current corticosteroid and immunosuppressive therapeutic approaches for nonspecific orbital inflammation.

Drugs	Mechanism of Action	Posology	Side Effects	References
Corticosteroids				
Prednisone	Glucocorticoid receptor-mediated cell signaling Suppression of polymorphonuclear leukocyte migration	Adults: 1 mg/kg Children: 1.0 to 1.5 mg/kg/day Reported total dose: 60 to 100 mg PO per day for one to two weeks, followed by a 5- to 6-week taper	Insomnia, hyperglycemia, weight gain, cataracts	[118,125,126,127]
Immunosuppressive agents				
Methotrexate	Inhibits dihydrofolate reductase Suppression of B- and T-cells	10 to 25 mg divided over 36 to 48 h every 1 to 4 weeks 15 to 25 mg per week for periods of 4 weeks to 36 months 12.50 mg per week with slow tapering	Mouth sores, nausea, abdominal pain, diarrhea, loss of appetite	[128,129,130,131]
Cyclosporine-A	Suppresses lymphocyte-mediated responses Inhibits T cell activation Decreases production of IL-1 and IL-2	Starting dose of 4 mg/kg/day and tapering to 2 mg/kg/day, continued for 18 months	High blood pressure, peripheral neuropathy, gingivitis	[132,133,134]

## 4. Emerging Molecular Approaches

The development of novel therapies for NSOI requires a thorough understanding of the biomolecular processes involved in its pathogenesis. Although many pathways are yet to be fully understood, novel approaches are being studied to improve NSOI diagnosis and management. These tools are engineered based on the CT imaging and MRI features of the disease, the histopathological features, and the biomolecular changes in patients, which are detected through peripheral blood analysis. By directly targeting these specific clinical and molecular changes, it is possible to leverage the challenges of current diagnostic and therapeutic tools by providing personalized medicine. 

### 4.1. Novel Diagnostic Tool

#### 4.1.1. Artificial Intelligence-Based Technologies 

NSOI diagnosis currently relies on imaging techniques (e.g., CT imaging and MRI), biopsy results, and laboratory investigations. Over the past years, a saliant interest towards deep learning and artificial intelligence has emerged for the diagnosis of orbital and eyelid diseases [135]. Orbital CT and MRI images are the mainstay in diagnosing orbital diseases, given the detailed analysis of anatomical structures and subsequent changes involved in the pathophysiology [136]. Using deep learning, in conjunction with the paraclinical data, we believe that more accurate discrimination of NSOI can be achieved (Figure 6). 

Deep learning refers to a type of representation learning model where learned features can be compositional or hierarchical [137]. Feature learning is approached by analyzing the simple features (e.g., image intensity, textures, and edges) encompassed in the more complex features (e.g., shape, lesions, organs, and adjacent structures) in order to map the compositional nature of the image and output an accurate classification that can be a clinical diagnosis as well as the presence or absence of a disease [137]. Convolutional neural networks (CNNs) are among the most commonly used deep learning models; they have fostered numerous revolutionizing research advances in the medical field, specifically in orbit imaging techniques [135,138,139,140,141]. CNNs are engineered to have multiple layers, where each is programmed to learn the various features of the given input (e.g., image). Ensuing each layer, a filter—called a kernel—allows to output a detailed and defined image [142]. Multiple deep learning techniques can be used during the image representation steps, such as image segmentation (i.e., locating the boundaries of an image in order to convert the segmented region into a region of pixels) and data augmentation (i.e., creating modified copies of the image data set by either rotating the image, altering its intensity, or flipping the image) [143]. Over time, it is expected that the use of deep learning technologies will lead to smart medicine, where diagnoses and treatments will be designed for each patient.

The concept of radiomics is defined by the extraction of image features from large amounts of highly complex and quantitative images and providing their subsequent analysis for proper classification [144]. In this context, the MRI-based radiomic signatures can be used to discriminate NSOI from other orbital pathologies. Guo et al. have demonstrated that radiomic features can be applied for the differentiation of NSOI from ocular adnexal lymphoma (OAL) by using the least absolute shrinkage and selection operator (LASSO) procedure [145]. The LASSO procedure, also known as LASSO regression, is used in the field of deep learning to outline associations and relationships between variables to output a prediction. By using five radiomic features, the team demonstrated discriminating OAL from NSOI with an area under the curve (AUC) of 0.74 [145]. More recently, by using the histopathological features of the biopsies as well as the MRI-derived images, in particular the T1-weighted image (T1WI), the T2-weighted image (T2WI), and the contrast-enhanced T1W1, Xie et al. were able to accurately differentiate OAL from NSOI [146]. They proceeded by extracting the multimodal radiomic features from the MRI sequences and validating with CNNs, which yielded an AUC for the diagnostic power of the DL model of 0.953 (95% CI, 0.895–1.000) [146]. Similarly, Hou et al. have demonstrated the effectiveness of bag-of-features (BOF)-based radiomics in discriminating OAL from NSOI, with an AUC of 0.803 (95% CI: 0.725–0.880) [147]. BOF and CNN are algorithms both based on the principle of extracting and learning the features of an image during the training step, thereby providing an accurate classification during the validation step [148]. BOF uses image features in an orderless collection, whereas CNN learns through hierarchical layers of representation [149]. The differences in the efficacy of discriminating NSOI from other ocular diseases using BOF and CNN may be explained by the higher performance of CNN during feature extraction for image classification, as well as the fine-tuning following each layer [149].

Clinically, these tools have a great advantage and potential for the time-sensitive diagnosis of NSOI. Delays in the diagnosis and treatment of NSOI can lead to an increased risk of complications, such as optic neuropathy and subsequent vision loss [150]. In this scenario, the possibility to use artificial intelligence-based imaging algorithms, in combination with the clinical information of the patient, can significantly lead to rapid and efficient management—a step towards smart medicine.

#### 4.1.2. miRNAs as a Diagnostic and Treatment Response Biomarker

miRNAs have been used for many years as credible diagnostic and treatment response markers in inflammatory conditions, such as inflammatory bowel disease [151], RA [152], and cancer [153]. miRNAs were also shown to be upregulated in idiopathic inflammatory myopathies, which translated to dysregulation in the IFN, anti-viral, and T-cell signaling pathways [154]. More recently, as previously discussed, miRNA clusters have been shown to be associated with NSOI; however, further investigation is required to better understand their disease-specific expression, sensibility, and specificity—parameters that have not yet been characterized [92]. Given the possibility to apply miRNA expression to disease diagnosis and treatment, over the past years, numerous miRNA quantification platforms have shown great accuracy and efficiency in quantifying miRNAs in biofluids, with small RNA sequencing (RNA-seq) having the highest accuracy with an AUC of 0.990 [155,156,157]. These tools are clinically crucial given their non-invasive nature, low cost, and high reliability. 

### 4.2. Novel Therapeutic Approaches 

#### 4.2.1. Biologic Agents 

The current gold standard treatment of NSOI involves the use of systemic corticosteroids, as mentioned in previous sections, along with NSAIDs and immunosuppressants (i.e., calcineurin inhibitors and antiproliferative drugs). Over the past years, the clinical interest towards personalized medicine has led to the development of new treatment options, such as biological agents (i.e., infliximab, rituximab, adalimumab, tocilizumab, and etanercept) for numerous autoimmune diseases, such as RA. Their off-label use is mainly important in the case of patients with refractory NSOI, given their high cost, possible adverse effects, and associated complications [38]. It is important to note that current knowledge on biologic agents for the treatment of NSOI relies on preclinical and clinical studies that were performed in models or patients who did not have NSOI, therefore rendering their use “off-label”. Over the past years, numerous case studies have reported the use of these biologic agents for the treatment of NSOI following board approbation which will be reviewed in this section (Table 3).

#### 4.2.2. Anti-TNFα Agents

The role of TNFα in autoimmune diseases and chronic inflammation has been thoroughly reviewed by Jang and colleagues [182]. Moreover, studies have also reported an upregulation in the expression of TNFα in NSOI [63,70]. With the emergence of personalized medicine, novel therapeutic approaches are aiming to halt TNFα expression upregulation.

TNFα, a homotrimer secreted by activated T cells, macrophages, and natural killer (NK cells), exerts its action by binding to its receptor TNFR1 or TNFR2, therefore triggering a series of inflammatory regulators such as cytokines and chemokines [183]. TNFα can be found in human cells in a soluble (sTNFα) or transmembrane (tmTNFα) form. It is initially synthesized as tmTNFα and subsequently converted to sTNFα through the action of the TNFα-converting enzyme (TACE) [184]. Both forms mediate their proinflammatory action by activating the signaling pathways of TNFR1 and TNFR2; however, tmTNFα mainly exerts its action through TNFR2 [185]. The activation of TNFR1 leads to the recruitment of the TNFR1-associated death domain (TRADD) and the subsequent activation of programmed cell death. During these cell death pathways, three distinct signaling complexes are formed: complex I, complex IIa, complex IIb, and complex IIc [182]. Through the recruitment of various molecular partners, each complex is involved in distinct cellular effects. Complex I leads to the activation of NFκB and MAPKs, which are involved in cell proliferation, inflammation, and cell survival. Complex IIa and IIb lead to the activation of caspase-8 and subsequent apoptosis, whereas complex IIc is involved in necroptosis and inflammation through the action of the mixed lineage kinase domain-like protein (MLKL). Conversely, TNFR2 signaling leads to the recruitment and association of TNFR-associated factor (TRAF), with subsequent recruitment of complex I and MAPKs, AKT, and NFκB signaling pathways [182]. Taken together, it is evident that inappropriate activation of TNFα signaling pathways can lead to excessive inflammation and disease development. TNFα inhibitors have been successfully developed to counter these pathological effects, such as the soluble TNFR2-Fc recombinant (Etanercept), the mouse-human chimera monoclonal antibody (Infliximab), and the human monoclonal antibody (Adalimumab) [158]. Etanercept contains an Fc-fusion protein of the extracellular domain of TNFR2, therefore its pharmacological effects are the result of TNFR2 antagonism. Conversely, adalimumab and infliximab encompass a 3:3 complex with TNFα and the Fab fragments [186]. This precise 3D orientation of adalimumab and infliximab allows the antibodies to bind with sTNFα and tmTNFα [186].

Over the past years, infliximab has shown great potential for treating orbital inflammation in patients with NSOI [159,160,161,162]. Similarly, a recent retrospective study on patients’ medical records from 2007 to 2016 has shown the efficacy of infliximab in controlling refractory orbital myositis [163]. Patients who were initiated on infliximab at 7.00 ± 6.83 mg/day showed a decrease in systemic corticosteroid need (absolute value decrease of 28.57 ± 14.35 mg/day) [163]. Moreover, six out of seven patients (85%) achieved long-term remission [163]. Adalimumab efficiency as a therapeutic option for orbital inflammation has also shown satisfactory results; however, overall efficiency was shown to be lower than infliximab. A retrospective study in Cambridge (Massachusetts) showed only a 43% 1-year remission ratio of adalimumab, whereas a long-term remission rate of 85% was achieved with infliximab [163,168]. Similarly to adalimumab, etanercept showed a lack of evidence regarding clinical remission in patients with inflammatory bowel diseases (IBD) [169]. A recent study has demonstrated a highly dynamic pharmacokinetic of infliximab in patients with autoimmune disorders; patients with Crohn’s disease exhibited a higher terminal elimination half-life of the infliximab-TNF complex in comparison to patients with RA and ankylosing spondylitis [164]. A potential source of anti-TNF agent efficiency in NSOI could therefore be explained by the variability in drug clearance. Furthermore, NFκB signaling pathway upregulation in patients with IBD was shown to negatively alter the anti-TNF agent response [165,166]. Hypothetically, patients with recurrent NSOI could therefore exhibit upregulated NFκB signaling that is resistant to treatment. Drug safety and innocuity are important aspects of drug development. Recent studies have thus investigated the impact of anti-TNF agents on the incidence of side effects and complications. Adalimumab was discontinued in several patients given the side effects, such as arthralgias and nausea [168]. Using the BIOGEAS Registry, Pérez-De-Lis et al. have investigated the incidence of autoimmune diseases with anti-TNF agents as well as high morbidity side effects [167]. CNS demyelination was reported in 0.33 cases per 1000 patients, as well as optic neuritis, in patients using etanercept and infliximab at most [167]. 

#### 4.2.3. B-Cell Modulating Agents

Rituximab is a humanized chimeric anti-CD20 monoclonal antibody whose primary function is to induce cell death in CD20+ cells [187]. CD20, a member of the membrane-spanning 4-domain family A (MS4A) protein family, is a surface protein expressed on B-cells [188]. It encompasses four hydrophobic transmembrane domains, two extracellular domains, and one intracellular domain. It is coupled to the CD40 protein, B-cell receptor (BCR), the major histocompatibility complex class II (MHCII), and the C-terminal src kinase-binding protein (CBP) [188]. CD20 was shown to be necessary for efficient BCR signaling in B cells [188]. As discussed previously, a self-maintained B cell expansion in NSOI could be one of the hallmarks of sustained inflammation. Therefore, the clinical approach consists of blocking B cell signaling pathways and inducing cell death to halt B cell expansion. 

Over the past year, numerous studies have reported the successful use of rituximab in refractory NSOI [170,171,172]. More recently, Ng et al. performed a comprehensive literature review and demonstrated the efficacy of rituximab for the treatment of non-infectious and non-malignant orbital inflammation [173]. A positive therapeutic response was observed in 88% of patients, with no side effects in 83.3% of participants [173]. Only 11% of participants showed a disease recurrence; sustained disease remission was observed in 88% of participants [173]. A recent case report regarding a 25-year-old female patient who presented with NSOI demonstrated that three cycles of a two-session per cycle rituximab infusion as first-line treatment (100 mg IV on days 1 and 15) were sufficient to obtain disease remission at 1-year post-treatment [174]. Conversely, in a 32-year-old patient with known psoriasis and systemic lupus erythematous (SLE) with daily prednisone treatment (equivalent to 10 mg), rituximab administration was shown to significantly improve the periorbital myositis clinically and on imaging [175]. Similar efficiency for rituximab was reported in another case report of SLE-associated NSOI [189]. The backbone of initiating rituximab in call cases of NSOI pins one’s hope on histology results; lymphocytic infiltrates shown on biopsies usher in the clinical decision to start anti-CD20 agents such as rituximab. 

Most of the preclinical studies regarding rituximab’s anti-inflammatory effects were studied in RA models [190]. Given that both RA and NSOI induce dysregulation in the immune system, one can generalize their effects to NSOI. In RA patients treated with rituximab (intravenous administration of 1 g on days 1 and 15), anti-CD20 therapy was shown to significantly reduce CD19+ B cells in peripheral blood [191]. By performing lymph node biopsies, Ramwadhdoebe et al. have demonstrated that rituximab leads to a concomitant decrease in T cell activation following B cell depletion [192]. When analyzing the biomolecular changes that ensued with rituximab treatment, it was shown that intraperitoneal injections of rituximab (250 mg/kg/week) downregulated the expression of NFκB, TNFα, and IL-6, as well as the GM-CSF signaling pathway, in a murine collagen-induced RA model [193].

#### 4.2.4. Interleukin-6 Receptor Binding Agents

The role of IL-6 in NSOI has been thoroughly covered in the previous sections, where it was shown to promote chronic inflammation and fibrosis [77,78,82]. Therefore, leveraging the inflammatory effects of IL-6 in NSOI with blocking agents such as tocilizumab is a novel therapeutic approach. Tocilizumab is a humanized monoclonal IgG1 antibody that inhibits the molecular actions of IL-6 by binding to its receptor (i.e., IL6R)—either the soluble or membrane-bound receptor [176,177]. Tocilizumab was shown to prevent the recurrence of NSOI in a 59-year-old patient with refractory NSOI over a span of 6 years when used at a maintenance dose of 4 mg/kg and following an initial treatment at a dose of 8 mg/kg [178]. However, in another case report, it was demonstrated that an initiation dose of 4 mg followed by 8 mg every 4 weeks to treat orbital pseudotumor couldn’t achieve inflammation control at 9 months [179]. These conflicting results outline the importance of treatment duration as well as the highly complex and heterogeneous entity of NSOI. In fact, in a case of unresponsive orbital myositis to systemic corticosteroid treatment, with subsequent unsatisfying clinical outcomes with immunosuppressants and other biological agents (i.e., infliximab and adalimumab), off-label use of tocilizumab at a starting dose of 8 mg/kg/day in combination with prednisone (60 mg/day) has shown great efficiency in stabilizing the clinical presentation of the disease while being able to further achieve tapering of prednisone following 9 cycles of treatment [180]. However, tocilizumab comes with its own burden of side effects. In patients with hematopoietic stem cell transplantation, single-dose administration of tocilizumab before the procedure was shown to be associated with significantly higher levels of depression scores at day 28 [181].

### 4.3. Plasmapheresis and Intravenous Immunoglobulin 

Plasmapheresis and intravenous Ig (IVIG) are two biomedical techniques involved in autoantibody removal by filtration and neutralization, respectively [194,195]. Given that most molecular dysregulations involved in NSOI affect the immune system and molecule expression within the peripheral blood, these technologies bring promising solutions for the treatment of NSOI. In concordance with this hypothesis, a case report successfully demonstrated the impact of plasmapheresis on NSOI control. Following five successful plasmapheresis sessions, preceded by 3 days of systemic corticosteroid, a 49-year-old female patient with NSOI exhibited clinically significant improvement of her symptoms without any recurrence at 9 days post-plasmapheresis therapy [196]. Conversely, it is suggested that IVIG therapy can mediate anti-inflammatory effects by activating the inhibitory Fc receptor pathway [197,198].

### 4.4. Intra-Orbital Drug Delivery

Lacrimal gland inflammation, also known as dacryoadenitis, is a common clinical presentation of localized NSOI [110]. In 20% of patients, bilateral dacryoadenitis can be observed in a simultaneous or sequential manner [110]. Furthermore, inflammatory dacryoadenitis due to NSOI may present with resistance to treatment and may require multiple treatment courses. In order to overcome these drawbacks, an ongoing clinical research trial is investigating the impact of direct lacrimal gland steroid delivery (i.e., intra-orbital steroid delivery) on the recurrence and duration of remission in patients with NSOI (ClinicalTrial.gov ID: NCT03958344). Although the study is currently in the recruitment phase, previous studies have demonstrated the efficacy of intra-orbital corticosteroid injection for the treatment of NSOI [126,199,200]. As mentioned in Section 2, disturbance in the homeostasis of the immune system marks the pathogenesis of NSOI, which is represented by enhanced T cell activation and recruitment to soft tissues. Direct anti-inflammatory drug delivery to inflamed tissues can significantly inhibit the pro-inflammatory processes and enhance tissue healing.

## 5. Conclusions

In conclusion, this review of NSOI underscores the intricate interplay of genetic, environmental, and immunological factors in its pathogenesis and highlights the pivotal role of molecular insights in advancing our understanding and management of this condition. The exploration of NSOI through a molecular lens has revealed complex signaling pathways, cytokines, and mediators that contribute to its development and progression. This knowledge has not only refined our diagnostic capabilities, but it has also opened new avenues for targeted therapies, including the use of biologics and immunomodulatory agents, offering a more personalized and effective approach to patient care.

As we continue to delve deeper into the molecular underpinnings of NSOI, we remain optimistic about the future of this disease. The understanding of its etiology, ranging from autoimmune and systemic diseases to environmental factors, complements the strides made in the molecular realm. This synergy of etiological and molecular insights not only improves patient care but also serves as a beacon of hope for addressing similar complex conditions. The advancements in understanding complex immune dysregulations, such as the roles of T cells, B cells, and various cytokine pathways in NSOI, are crucial in this endeavor. The novel discoveries and advancements in the field of deep learning enhance the performance of artificial intelligence models in multi-input diagnosis. This journey of discovery and innovation, rooted in a multidisciplinary approach, is a testament to the relentless pursuit of excellence in molecular science that eventually translates to patient care. It inspires confidence that, with ongoing research and collaboration, we will continue to unlock new frontiers in the fight against challenging idiopathic conditions like NSOI that have no cure. Ultimately, this paves the path toward a future where the potential to significantly mitigate or even eradicate the visual impairments and aesthetic concerns associated with orbital diseases becomes a tangible reality. 

## Figures and Tables

**Figure 1 ijms-25-01553-f001:**
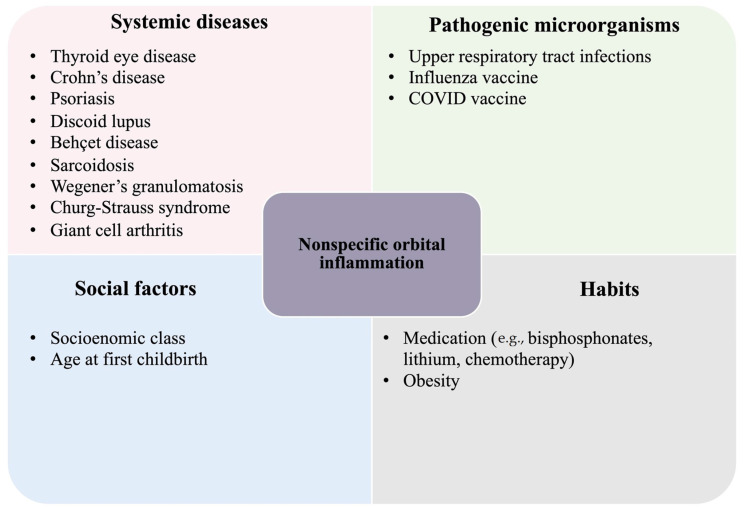
Etiologies associated with nonspecific orbital inflammation. Nonspecific orbital inflammation (NSOI) was shown to be linked to diverse conditions ranging from systemic diseases to diseases induced by pathogenic microorganisms, social factors, and personal habits.

**Figure 2 ijms-25-01553-f002:**
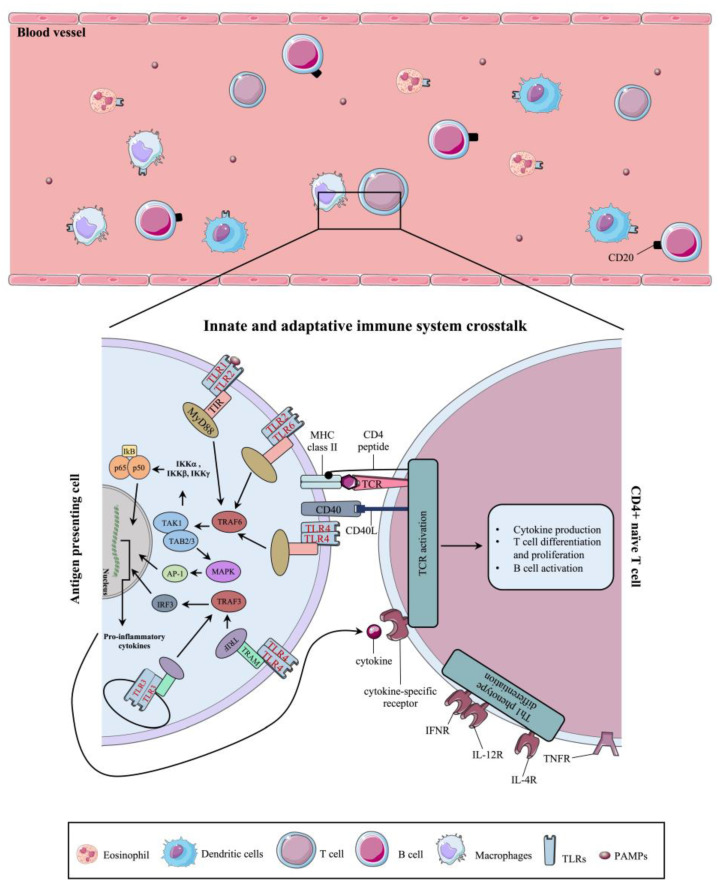
Schematic representation of the innate and adaptative immune systems. The innate immune system encompasses neutrophils, eosinophils, antigen-presenting cells (APCs), such as macrophages and dendritic cells (DCs), and natural killer (NK) cells. Activation of the innate immune system is achievable through the recognition of soluble pathogen-associated molecular pattern molecules (PAMPs) by pattern recognition receptors (PRRs), such as toll-like receptors (TLRs), on innate immune cells. TLR-ligand binding induces downstream signaling pathway activation, which ultimately leads to the production of proinflammatory cytokines. APCs constitute the bridge between the innate and adaptive immune systems. APCs activate T cells through the binding of CD40 with the CD40 ligand, major histocompatibility class (MHC) II binding with the T cell receptor, and cytokine-specific receptor stimulation. T cell differentiation is ultimately regulated by the nature of the given cytokine. The figure was partly generated using Servier Medical Art provided by Servier and licensed under a Creative Commons Attribution 3.0 Unported license.

**Figure 3 ijms-25-01553-f003:**
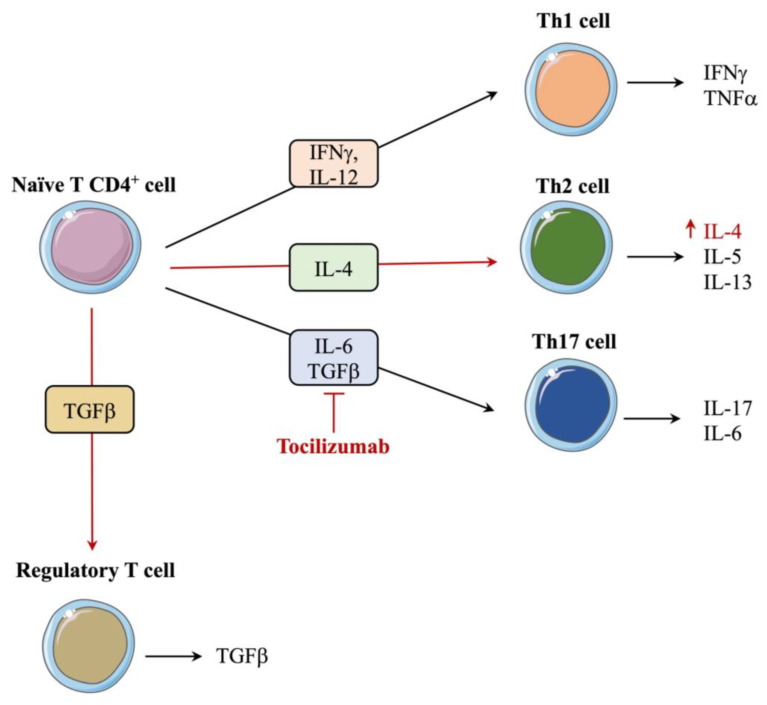
Schematic representation of T cell lineage differentiation and modifications involved in nonspecific orbital inflammation. Cytokine production in the microenvironment mediates T cell lineage differentiation into specific subtypes, which in turn induces specific cytokine production. In nonspecific orbital inflammation (NSOI), IL-4 production was shown to be enhanced, a hallmark of the Th2 cell lineage. Furthermore, given the dysregulation in IL-6 levels, which are involved in Th17 phenotypes, tocilizumab—an anti-IL6 agent—can be used for the treatment of NSOI. Dysregulations and drugs involved in NSOI are represented with red arrows or in bold format. The figure was partly generated using Servier Medical Art provided by Servier and licensed under a Creative Commons Attribution 3.0 Unported license.

**Figure 4 ijms-25-01553-f004:**
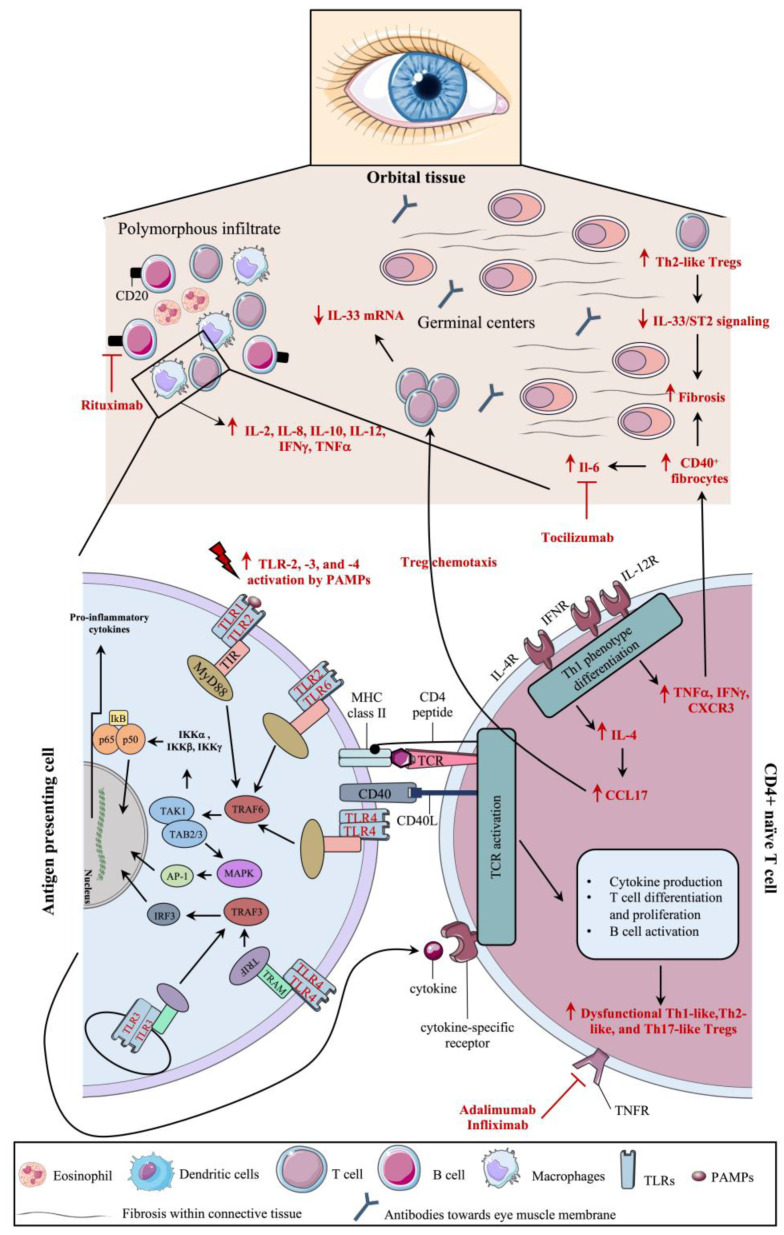
Schematic representation of molecular pathways hypothesized to be involved in the pathogenesis of nonspecific orbital inflammation. Nonspecific orbital inflammation (NSOI) is known to be induced by dysregulation in the immune system, which is outlined in this figure. NSOI-induced dysregulations are represented in bold red and with red arrows. The figure was partly generated using Servier Medical Art provided by Servier and licensed under a Creative Commons Attribution 3.0 Unported license.

**Figure 5 ijms-25-01553-f005:**
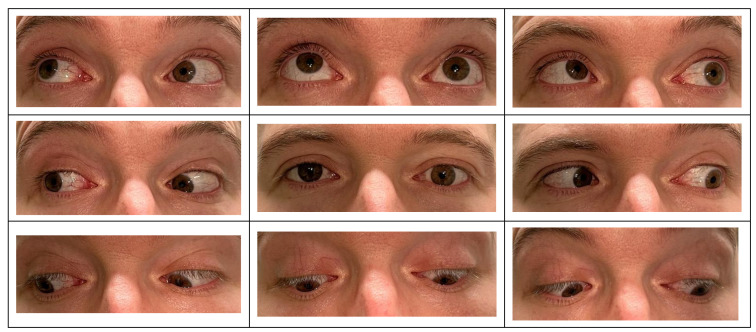
Ocular motility examination in orbital myositis due to nonspecific orbital inflammation. This color photograph series illustrates an ocular motility examination of a patient with orbital myositis as a secondary condition to NSOI in the left eye. The images showcase the primary, secondary, and tertiary positions of gaze, documenting the eye’s alignment and movement limitations in each position. Copyright © Dr. Patrick Daigle and Dr. Kevin Yang Wu. All rights reserved. No part of this image may be reproduced or transmitted in any form or by any means without prior written permission from the copyright holders.

**Figure 6 ijms-25-01553-f006:**
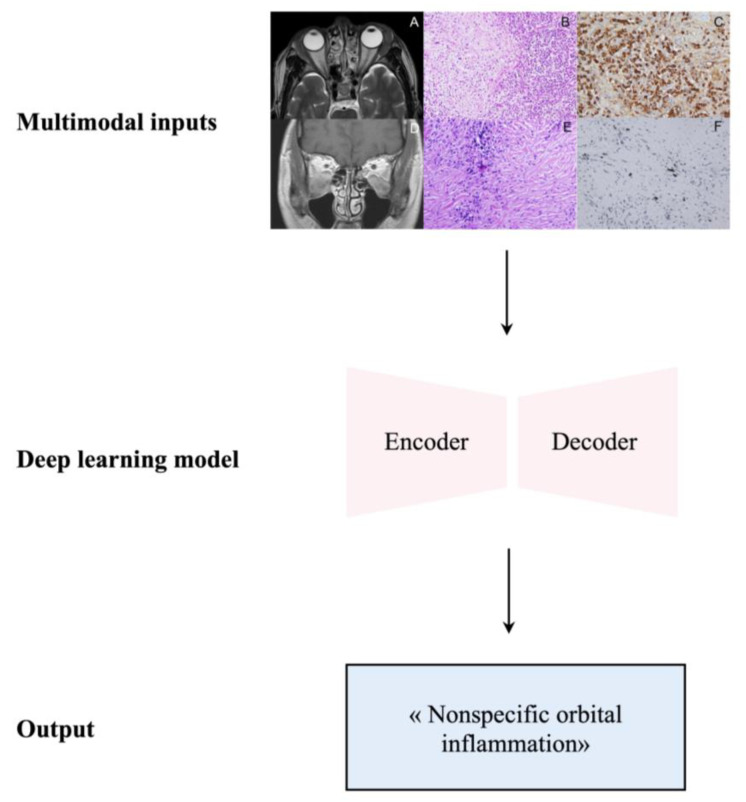
Overview of deep learning applications for connecting images to texts. The use of deep learning in the diagnosis of nonspecific orbital inflammation (NSOI) has shown great success. By using multimodal inputs (i.e., imaging results, such as CT orbits and orbit MRIs, or histopathology results from biopsies), the architecture can output the correct diagnosis, therefore leveraging the challenges associated with the non-accurate diagnosis of NSOI. Multimodal inputs can be (**A**) T2-weighted MRI image of lacrimal glands, (**B**) histopathological slide showing lymphoplasmacytic infiltration of a lacrimal gland, (**C**) positive IgG4 immunostaining, (**D**) T1 enhanced MRI image of the orbits, (**E**) histopathological slide with lymphocytic infiltration, or (**F**) negative IgG4 immunostaining. Parts of this figure (multimodal inputs) were reprinted from Non-specific orbital inflammation: Current understanding and unmet needs, 81, Lee et al. 100885 [1], Copyright (2021), with permission from Elsevier.

**Table 1 ijms-25-01553-t001:** Summary of a miRNA cluster potentially involved in nonspecific orbital inflammation pathogenesis ^a^.

Identified miRNA	Known Function in Inflammation ^b^	References
miR-223-3p, miR-223-5p	Upregulated in granulocytes, macrophages, and T cells. Upregulated by NFκB. Proliferation and differentiation of granulocytes and macrophages. Induces Treg differentiation in vivo.	[93,94,95]
miR-148a	Regulates monocyte-derived DCs and attenuates psoriasis-induced inflammation progression. Impairs B cell tolerance and induces autoimmunity through *Gadd45a*, *Pten*, and *Bcl2l11* suppression.	[96,97,98]
miR-365-3p	Negatively regulates IL-17 through ARRB2 targeting in a murine asthmatic model. Enhances apoptosis and inhibits cell proliferation by IL-1β and IL-6 downregulation in synoviocytes of mice with rheumatoid arthritis.	[99,100]
miRNA-140	Enhances cell death through apoptosis.	[101]
miR-193a	Promotes granulocyte differentiation.	[102]
miR-29a-3p and U6 snRNA	To be investigated in autoimmunity.	

^a^ miRNA cluster identified by OpenArray miRNA profiling in patients with nonspecific orbital inflammation. Laban et al. (2020) [92]. ^b^ The reported functions are based on preclinical studies regarding diverse autoimmune diseases, excluding nonspecific orbital inflammation, given the lack of data.

**Table 3 ijms-25-01553-t003:** Summary of off-label biological agents used in the treatment of nonspecific orbital inflammation.

Drug	Structure	Studies	Side Effects ^a^	References
Anti-TNF agents				
Infliximab	Mouse-human chimera monoclonal antibody	Great efficacy in treating NSOI. Induced a decrease in systemic corticosteroid need at a dose of 7.00 ± 6.83 mg/day. Long-term remission achieved in 85% of patients with NSOI.	Dynamic pharmacokinetics with variable terminal half-life times. Mostly associated with optic neuritis. Negatively influenced by NFκB upregulation.	[158,159,160,161,162,163,164,165,166,167]
Adalimumab	Human monoclonal antibody	Lower efficacy: 1-year remission observed in 43% of patients with NSOI.	Associated with higher levels of arthralgia and nausea. Negatively influenced by NFκB upregulation.	[158,163,165,166,168]
Etanercept	Soluble TNFR2-Fc recombinant protein	Lack of evidence regarding clinical remission in patients with IBD.	Mostly associated with optic neuritis. Negatively influenced by NFκB upregulation.	[158,165,166,167,169]
B-cell modulating agent (anti-CD20)				
Rituximab	Humanized chimeric anti-CD20 monoclonal antibody	Successful efficacy in refractory NSOI. Post-therapeutic response in 88% of patients, with the absence of side effects in 83% of participants. Disease remission at 1 year achievable with IV infusion of 100 mg on days 1 and 15.	Very few side effects. Disease recurrence in 11% of patients.	[170,171,172,173,174,175]
Anti-IL-6 receptor agents				
Tocilizumab	Humanized monoclonal IgG1 antibody	Prevents NSOI recurrence over 6 years, at least at a maintenance dose of 4 mg/kg. Uncontrolled inflammation at 9 months with 8 mg every 4 weeks.	Associated with higher levels of depression.	[176,177,178,179,180,181]

^a^ Side effects encompass those outlined in various autoimmune and inflammatory diseases. Abbreviations: NSOI, nonspecific orbital inflammation; IBD, inflammatory bowel disease.

## Data Availability

Not applicable.

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
