# Peer review of "Nonspecific Orbital Inflammation (NSOI): Unraveling the Molecular Pathogenesis, Diagnostic Modalities, and Therapeutic Interventions"

_ijms, 2024, doi:10.3390/ijms25031553_

Round 1

Reviewer 1 Report

Comments and Suggestions for Authors

In this manuscript, authors have reviewed nonspecific orbital inflammation presenting its etiology, diagnosis, and therapeutic treatment approaches. Overall, it has been very detailed, well-constructed and covers all aspects that is of interest to the readers. I have following minor concerns.

It would have been more compelling to provide photographs of patients suffering from different types of nonspecific orbital inflammation with clinical presentations.

Viral infections are considered as one of the prominent factors in the pathogenesis of NSOI. This can be further detailed in a separate subsection, especially elaborating on EBV.

Diverse immunosuppressants are routinely studied in the treatment of NSOI such as cyclosporine-A and methotrexate, mostly replacing corticosteroids. These can be added to the existing table 2.

Author Response

Dear Reviewer,

We are grateful for your valuable insights and suggestions regarding our manuscript on nonspecific orbital inflammation (NSOI). Your feedback has greatly contributed to enhancing the quality and depth of our work. We are pleased to inform you that we have addressed each of your concerns in our updated manuscript as follows:

Inclusion of Patient Photographs: In response to your suggestion, we have included a series of photographs that depict clinical presentation of NSOI. (figure 5)

Expanded Section on Viral Infections and EBV: We agree with your observation regarding the significance of viral infections, particularly Epstein-Barr Virus (EBV), in the pathogenesis of NSOI. Consequently, we have added a dedicated subsection that delves into the role of viral infections, with an emphasis on EBV. This subsection provides a detailed overview of the current understanding of viral etiologies in NSOI, supported by recent studies and clinical findings.

Updated Table on Immunosuppressants: Recognizing the importance of the evolving therapeutic landscape in NSOI management, we have revised Table 2 to include additional information on immunosuppressants like cyclosporine-A and methotrexate. This updated table offers a comprehensive comparison of these drugs alongside corticosteroids, highlighting their usage, efficacy, and side effect profiles based on the latest research.

We sincerely appreciate your constructive comments, which have been instrumental in refining our manuscript. We believe that these enhancements will make our article more informative and valuable for readers in the field of ophthalmology. Thank you for your thoughtful review and the opportunity to improve our work.

Warm regards,

Reviewer 2 Report

Comments and Suggestions for Authors

Manuscript is well prepared. Only minor corrections are needed.

Author Response

Dear Reviewer,

Thank you for your thorough review and constructive feedback on our manuscript regarding nonspecific orbital inflammation (NSOI). We greatly appreciate your comments, which have provided us with an invaluable opportunity to enhance the manuscript. Please find below how we have addressed each of your major and minor points:

Major Comments:

Abstract Revision: We have revised the abstract to better highlight the novelty of our manuscript, emphasizing the unique contributions and perspectives it brings to the current understanding of NSOI.

Keywords Update: Acknowledging your observation, we have updated the keywords to more accurately reflect the manuscript's content. We have included and removed terms that are more or less relevant to the core topics discussed.

Clinical Characteristics of NSOI: We have included a comprehensive section detailing the clinical characteristics of NSOI. This section provides a thorough overview of the clinical presentations, enhancing the manuscript's diagnostic focus. We have also included clinical images to highlight the clinical manifestation of this disease. (Figure 5)

Occurrence of NSOI in Introduction: The introduction now briefly mentions the occurrence of NSOI, providing readers with a contextual understanding of its prevalence and significance in clinical practice.

Inclusion of Figures from Other Articles: Since we have decided to include our own clinical images to highlight the clinical manifestation, using other research team’s clinical images would not be necessary – avoiding time to get copyright approval from these authors. (Figure 5)

Future Prospects Section: A new section on future prospects has been added toward the end of this manuscript, outlining how the information presented in our manuscript can contribute to improving the diagnosis and treatment of NSOI. This section offers insights into potential research directions and clinical applications.

Minor Comments:

Abbreviations List: A list of abbreviations used throughout the manuscript has been added at the end for easy reference.

Keywords Adjustment: We have removed “ADs and NSOI” from the keywords to ensure greater relevance and clarity.

Text Formatting Correction: The italics on “et al.” in line 123 have been removed, aligning with proper formatting standards.

Closing Remark:

We are grateful for your positive remarks about the overall preparation and structure of our manuscript. Your detailed review has been instrumental in guiding our revisions, and we believe these changes substantially enhance the manuscript's contribution to the field of ophthalmology.

Thank you once again for your valuable feedback and guidance.

Warm regards,

Reviewer 3 Report

Comments and Suggestions for Authors

The review article is original and very interesting.

The topic is very relevant, the aim of this review being to offer a comprehensive overview of Nonspecific orbital inflammation (NSOI) on its etiopathogenesis, clinical presentation, diagnostic methods, and management strategies. The article delves into the underpinnings of NSOI, examining immunological and environmental factors alongside intricate molecular mechanisms involving signaling pathways, cytokines, and mediators.

Methodology is very good, authors summarizing the research findings on emerging molecular discoveries and approaches, highlighting the significance of understanding molecular mechanisms in NSOI for the development of novel diagnostic and therapeutic tools. Various diagnostic modalities are scrutinized for their utility and limitations. Therapeutic interventions encompass medical treatments with corticosteroids and immunomodulatory agents, all discussed in light of current molecular understanding.

The results have summarized the diagnostic and therapeutic approaches for Nonspecific Orbital Inflammation.

The conclusions are not very consistent with the evidence and arguments presented, being rather narative.

The references are very relevant, including also some relevant author’s previous experience in the field.

I suggest some corrections

1.     The Abstract could be more detailed, to evidentiate the main findings of the article

2.     Introduction could be more detailed. Paragraphs from lines 44-49 are an unusefull story about the article

3.     Conclusions should also evidentiate the main findings of the article

Small correcctions

1.     There is an inconsistency in abbreviation of Nonspecific orbital inflammation (NSOI) or IOI. Should be uniformized in all the manuscript

4.     Some References (4, 70) do not follow the Authors Guide for MDPI Journals.

Author Response

Dear Reviewer,

We express our deepest gratitude for your thoughtful and constructive critique of our manuscript on nonspecific orbital inflammation (NSOI). Your appreciation of the originality and relevance of our work is highly encouraging. Following your invaluable suggestions, we have meticulously updated our manuscript to reflect the necessary amendments:

  1. Abstract Enhancement: We have expanded the abstract to more accurately convey the main findings of our review, thereby providing a clearer snapshot of the article's significant contributions to the understanding of NSOI.
  2. Introduction Revision: Acknowledging your feedback, we have restructured the introduction to eliminate unnecessary narrative from lines 44-49, focusing on a more detailed and substantive presentation of the topic.
  3. Consistency in Conclusions: The conclusions section has been carefully revised to ensure that it is consistent with the evidence and arguments presented throughout the article. We have made sure to underscore the main findings, reflecting a solid summary of our comprehensive review.

Small Corrections:

  1. Abbreviation Consistency: We have standardized the abbreviation of nonspecific orbital inflammation throughout the manuscript to consistently use "NSOI," rectifying the previous inconsistency with the term "IOI."
  2. Reference Formatting: References have been corrected to adhere strictly to the Authors Guide for MDPI Journals, ensuring that all citations meet the journal’s standards.

We are confident that these corrections have significantly improved our manuscript, aligning it more closely with the high standards expected by your esteemed journal. Your feedback has been pivotal in this revision process, and for that, we are sincerely appreciative.

Thank you for your guidance and the opportunity to enhance our work further.

Warmest regards,

Round 2

Reviewer 3 Report

Comments and Suggestions for Authors

The authors have made all the corrections suggested. I recommend the acceptance of the article in this revised 2 form